# Healthcare professionals' perspectives of the provision of, and challenges for, eating, drinking and psychological support post stroke: findings from semistructured interviews across India

Stephanie P Jones [ORCID],[1] Ranjit J Injety [ORCID],[2] Jeyaraj D Pandian [ORCID],[2] Sanjali Ratra,[2] PN Sylaja,[3] Veena Babu,[3] MV Padma Srivastava,[4] Sakshi Sharma,[4] Sudhir Sharma,[5] Jemin Webster [ORCID],[6] Amrit Koirala,[6] Pawna Kaushal,[5] Girish Baburao Kulkarni,[7] Anand Dixit,[7] Arvind Sharma,[8] Jagruti Prajapati,[8] Jo Catherine Weldon [ORCID],[1] Jennifer A Kuroski,[1] Caroline Leigh Watkins,[1] Catherine Elizabeth Lightbody,[1] on behalf of the NIHR Global Health Research Group on IMPROVIng Stroke CarE in India (IMPROVISE) Collaboration

For numbered affiliations see end of article.

**Correspondence to**
Dr Stephanie P Jones;
sjones10@uclan.ac.uk

## ABSTRACT

**Aim** This qualitative study explores with health professionals the provision of, and challenges for, postdischarge stroke care, focussing on eating, drinking and psychological support across India.

**Design** Qualitative semistructured interviews.

**Setting** Seven geographically diverse hospitals taking part in a Global Health Research Programme on Improving Stroke Care in India.

**Participants** A purposive sample of healthcare professionals with current experience of working with patients who had a stroke.

**Results** Interviews with 66 healthcare professionals (23 nurses (14 staff nurses; 7 senior nurse officers; 1 intensive care unit nurse; 1 palliative care nurse)); 16 doctors (10 neurologists; 6 physicians); 10 physiotherapists; 5 speech and language therapists; 4 occupational therapists; 4 dieticians; 2 psychiatrists; and 2 social workers resulted in three main themes: integrated inpatient discharge care planning processes; postdischarge patient and caregiver role and challenges; patient and caregiver engagement post discharge.

**Conclusions** Discharge planning was integrated and customised, although resources were limited in some sites. Task shifting compensated for a lack of specialists but was limited by staff education and training. Caregivers faced challenges in accessing and providing postdischarge care. Postdischarge care was mainly hospital based, supported by teleservices, especially for rural populations. Further research is needed to understand postdischarge care provision and the needs of stroke survivors and their caregivers.

## BACKGROUND

Stroke is a significant global health problem and a major cause of mortality and morbidity in high-income countries (HICs), and

## STRENGTHS AND LIMITATIONS OF THIS STUDY

⇒ We did not seek to interview healthcare professionals (HCPs) based in the community as services varied between the included sites and were not stroke specific.

⇒ Due to the diversity of the included hospitals, we were unable to make comparisons across services.

⇒ It is acknowledged that one in every three patients with stroke do not access appropriate healthcare due to non-affordability, usage of alternative medicines and difficulty in conveyance to hospital.

⇒ Participants represented a broad range of stroke services; of which, over a third were nurses of varying seniority.

⇒ Interview methods (telephonic or face-to-face) and interviews in local languages maximised participant engagement, and also enabled adherence to COVID-19 protocols, and inclusion of participants under COVID-19 restrictions at the point of interview.

increasingly in low-middle income countries (LMICs).[1] Seventy per cent of strokes occur in LMICs; the subsequent disease burden is greater than in HICs.[2] Life expectancy in India has recently increased to over 60 years of age,[3 4] leading to an increase in age-related, non-communicable diseases including stroke.[5 6] In India, the incidence of stroke ranges between 105 and 152/100 000 people per year.[7] Increasing incidence of stroke is observed among younger groups, with first ever strokes occurring around the age of 57 years,[8–10] increasing families' burden.

Stroke survivors often experience many disabilities including hemiparesis,

cognitive and communication problems, incontinence, dysphagia and psychological trauma,[11] having serious economic and social impacts.[12] In India, the majority of stroke survivors have disabilities, and the burden of long-term care often falls on family members.[9] This is particularly challenging when stroke survivors leave hospital with complex needs; requiring nasogastric (NG) tubes, catheters and support with communication and psychological problems.

Stroke rehabilitation has been found to reduce mortality, decrease institutionalised care[13] and increase functional outcomes.[14 15] Despite the known benefits, India has limited resources to provide postdischarge stroke care.[16] Accessing rehabilitation services is a challenge for many stroke survivors in India. There are enduring barriers including long delays in accessing rehabilitation services, poor transport between home and hospital[17 18] and a lack of trained personnel.[9] There are a lack of data about postdischarge stroke services and the support available for patients and caregivers.[14]

The resources for poststroke rehabilitation, especially the therapy workforce and infrastructure, are very limited in many LMICs.[19] Beyond urban tertiary centres, rehabilitation services are scarcely available. There are <10 skilled rehabilitation practitioners per 1 million people in many LMICs, which does not allow specialisation to treat a particular condition or disorders.[20] In India, it estimated that there are only 41 567 physiotherapists (PTs), 3100 occupational therapists (OTs) and 1700 speech and language therapists (SLTs), and these are geographically concentrated in urban areas leaving much of the nation without resources.[15] Given the scarcity of resources, the rehabilitation needs of the stroke survivors, especially in LMICs, remain largely unmet resulting in a growing need for cost-effective, multidisciplinary rehabilitation services, particularly postdischarge.[21]

This qualitative study aimed to explore with health professionals the provision of, and challenges for, postdischarge stroke focussing on eating, drinking and psychological support across India.

## Design
Qualitative semistructured interviews.

## Setting and subjects
Interviews were conducted with a purposive sample of healthcare professionals (HCPs) (doctors, nurses, PTs, OTs, SLTs, dieticians, psychiatrists and social workers (SWs)) with current experience of working with patients who had a stroke at seven geographically diverse hospitals that were a mix of private, government and teaching, and all taking part in a Global Health Research Programme on Improving Stroke Care in India (All India Institute of Medical Sciences, New Delhi; Baptist Christian Hospital, Tezpur; Christian Medical College, Ludhiana; Indira Gandhi Medical College & Hospital, Shimla; National Institute of Mental Health and Neurosciences, Bangalore;

Sree Chitra Tirunal Institute of Medical Sciences, Trivandrum; Zydus Hospital, Ahmedabad) between July and December 2021.

## METHODS
Interviews explored HCPs' views of stroke patient management in preparation for discharge and postdischarge support for patients and caregivers, focussing on eating, drinking and psychological support.

The interview guide (online supplemental file 1) was informed by published research and developed collaboratively by the research team. While not formally piloted, the interview guide was revised following feedback from research associates (RAs) and principal investigators at the included hospitals, to ensure that questions could be translated into regional languages (Assamese, Gujarati, Hindi and Malayalam). According to participant preference, interviews were conducted in English or regional languages, and took place either face-to-face or by telephone (dependent on current COVID-19 restrictions).

### Data collection
All interviews were digitally recorded, translated (as required) and transcribed verbatim. All transcriptions were verified by RAs prior to analysis. Data collection continued until saturation, determined as no new information provided to enable the generation of additional codes from across the participants recruited.

### Patient and public involvement
Patients and the public were involved in discussions to agree the priority areas that were the focus of postdischarge support interviews with HCPs (eating, drinking and psychological support).

### Analysis
Each transcript was read in full, and data explored using inductive coding and analysed using a reflexive thematic method. Thirty interviews were coded as a group and 36 by individual project members. Confirmative and contradictory results were discussed by the project team for consensus. Data were analysed thematically using NVivo software (V.1.5.2.). This manuscript was prepared in accordance with the Consolidated Criteria for Reporting Qualitative Research guidelines[22] (online supplemental file 2).

### Reflexivity
All interviews were undertaken by RAs (AD, AK, JP, PK, SR, SS, VB) who were known by participants, but were not someone who they worked with clinically and lasted between 15 and 47 min. Interviews were conducted in Assamese (n=1), Gujarati (n=1), Hindi (n=12), Malayalam, (n=3) and English (n=51). All members of the project team were involved in data analysis. Dependability of the data may vary based on any staffing changes following the research undertaken, and all interviews took place during the COVID-19 pandemic. Interviews

were undertaken with participants from seven hospitals across geographically diverse areas. To support the analysis, a supplementary mapping exercise was undertaken to identify hospital and community-based services. The findings from the mapping activity enabled cross validation of the interview findings.

## Ethical considerations

Ethics approval (REF: 2020-9715) was granted by the Health Ministry's Screening Committee (HMSC), India and each hospital's ethics committee. The study conforms to ethical standards of the Indian Council of Medical Research Guidelines.[23] In the UK, ethical approval was granted by the Science, Technology, Engineering and Medicine Ethics Committee, University of Central Lancashire (STEM 939). Ethical approval was granted for both written and/or verbal consent, which was obtained from all participants prior to any interviews.

## RESULTS

Sixty-six HCPs participated, 39 (59%) female, representing a range of health professionals (table 1).

Data saturation was not reached until 66 interviews had been completed and this likely reflects the diversity of the participants, hospital type, hospital settings and patient populations served.

Three themes were identified:
1. Integrated inpatient discharge care planning processes (five subthemes):
    1.1. Healthcare system and resources.
    1.2. Staff identification, and expectations of caregivers.
    1.3. Task-shifting.
    1.4. Patient and caregiver training.
    1.5. Education and support in preparation for discharge.
2. Postdischarge patient and caregiver roles and challenges (three subthemes):
    2.1. Patient and caregiver characteristics.
    2.2. Caregiver roles.
    2.3. Caregiver challenges.
3. Patient and caregiver engagement postdischarge (two subthemes):
    3.1. Postdischarge support—hospital.
    3.2. Postdischarge support—community based.

## Theme 1: integrated discharge care planning processes

Discharge planning processes were integrated throughout the patient pathway, beginning soon after admission although varied across sites. Discharge planning was frequently dependent on the availability of specialist staff, resources and facilities, and often resulted in task shifting (defined as the rational redistribution of tasks among health workforce teams) due to a lack of staff availability/expertise. Participants reported that HCPs provided patients and caregivers with tailored, targeted education, training and support throughout hospitalisation.

### Subtheme 1.1: healthcare system and resources

All sites had localised, inpatient discharge planning processes or protocols in place, but this was largely undocumented.

> We have discharge processes that the team work to but we don't have any written protocol but we are observing the patient from the admission. (01-05)

Staff identified that existing postdischarge management systems could be improved by formal documentation to facilitate a more consistent and comprehensive approach.

> I really feel like there is dire need of formulating a kind of discharge protocol for the patient. (03-02)

### Subtheme 1.2: staff identification, and expectations of caregivers

Identified by the clinical team, caregivers would most often be the spouse, another immediate or extended family member.

> We will ask about the immediate relatives, either a wife or husband, brother or sister, son or daughter. We also identify the immediate family or caregiver. (06-04)

Training took place with patients and caregivers, expecting that caregivers would primarily support and provide postdischarge care.

> They need to know how to give physiotherapy, occupational therapy, mental support, how to give food, speech related, so, they need to be a doctor, they need to be therapist, they need to be a nurse… If the caregiver is not up to the mark, the rehabilitation fails. (06-01)

On admission, an individualised approach was used to identify the most appropriate caregiver(s) for tailored training and support.

**Table 1** Participant demographics

| Healthcare professionals | N (%) |
| --- | --- |
| **Nurses** | **23 (35)** |
| Staff nurses | 14 (21) |
| Senior nurse officers | 7 (11) |
| Intensive care nurse | 1 (1.5) |
| Palliative care nurse | 1 (1.5) |
| **Doctors** | **16 (24)** |
| Neurologists | 10 (15) |
| Physicians | 6 (9) |
| Physiotherapists | 10 (15) |
| Speech and language therapists | 5 (8) |
| Occupational therapists | 4 (6) |
| Dieticians | 4 (6) |
| Psychiatrists | 2 (3) |
| Social workers | 2 (3) |

It depends on the status of the patient according to the medical and surgical condition of patient… We educate the relatives and patients as well as possible. (07-03)

In addition to, or in the absence of, family caregivers, some patients employed the services of professional caregivers.

Some patients have the capacity of paying, so they hire the physiotherapists, and they call them at their home…Some patients cannot pay but they need physiotherapy. (04-04)

### Subtheme 1.3: task shifting

While some sites had well-resourced stroke teams, there was a shortage of dieticians, SWs and therapists (PTs, OTs, SLTs, psychologists) at other sites, where existing staff, predominantly nurses, provided information and support.

At the time of discharge being a nurse, as we do not have speech and language therapist, then we [nurses] try to put patients in the upright position and ask them to speak as much as they can. (04-02)

We are at such a place where we do not get any psychologists, that thing [psychological problems] we [nurses] have to manage ourselves because we are very close to the patients. (04-04)

Staff often described task shifting as integral to their professional roles and responsibilities; however, the majority had not undertaken any formal training and had developed knowledge and skills largely through experiential learning.

No, we didn't get any specific training for this [providing dietary advice]. Whatever we have studied during our nursing training, learned from doctors, and from the internet, on that basis we provide this information. (04-01)

### Subtheme 1.4: patient and caregiver training

Patient and caregiver training usually began at admission and continued until discharge, with staff encouraging caregivers to observe all aspects of patient care, as required, including the preparation and administration of medication, positioning, changing dressings, feeding, the management of NG tubes, maintaining hygiene and rehabilitation exercises.

We make sure the person who will be taking care of the patient at home is there from the beginning, so that they are very much part of all the processes that happen in the ward: changing dressings, changing position, and feeding and all those things. (02-06)

Staff felt that patients and caregivers should be prepared both mentally and physically for discharge but shared few examples of any formal psychological support or preparation.

We tell them, start observing from now, because when you go home, you have to do it by yourself. So, they also get ready mentally and physically. (02-02)

### Subtheme 1.5: education and support in preparation for discharge

To prepare the patient and caregiver for discharge, training delivered by HCPs was predominantly provided through verbal instruction and explanation.

When the relatives are in the hospital, we teach them verbally and some of the relatives they themselves write down [instructions]. (02-01)

Training was often supplemented by the provision of information, in a variety of formats including leaflets, videos, written instructions, diet charts and discharge summaries. Some PTs provided prerecorded videos of exercises, available through websites such as YouTube and/or recorded patients undertaking exercises.

But then we do use videos like I said, sometimes pictures of exercises, but it's not uniform for all. (03-05)

While training took place throughout patients' hospitalisation, education and information was largely provided within 24–48 hours of discharge. Leaflets were available in some sites, in most local languages, covering a range of topics to support the caregiver. In other sites, discharge summaries and diet charts were the only source of written information and instruction.

In our stroke unit, we have made pamphlets regarding safe swallow techniques and how to feed the patient. (06-02)

Accessible written information in appropriate languages was viewed as a valuable source of information, enabling information to be passed from the patient or primary caregiver in hospital to the main caregiver at home.

We have pamphlets which are given to the patient depending on the language of their preference… that helps in dispersing information further because the caregiver who is there with the patient in the hospital may not be the same caregiver who is there at home. (01-04)

### Theme 2: postdischarge patient and caregiver roles and challenges (three subthemes)

The patient's stroke and patients'/caregivers' levels of literacy were viewed as important factors in the subsequent 'successes' of postdischarge rehabilitation. HCPs identified multiple challenges in providing education and training, and specific challenges for the patients and caregivers themselves.

### Subtheme 2.1: patient and caregiver characteristics

Female, family members were often the preferred caregivers for both male and female patients.

We prefer training the female relative because the boys or males they usually go out for work. (02-01)

Stroke severity, post-stroke complications, cognitive and communication problems influenced the complexity and challenges of education and training for patients and their caregivers.

Some patients having pressure sores, second or third degree, then it will be really difficult to handle the wound, the relatives need more information about how to care for the wound, how to reduce the severity, when to change dressings, and how much time between changing the patient's position. (07-02)

Patients' and caregivers' educational attainments were viewed as important in effective training delivery. Patients and caregivers from rural locations often had lower levels of literacy that impacted their ability to understand, and to pay for postdischarge care needed.

Those who come from village they are not educated and are daily wage workers who are below the poverty-line… they really cannot understand the importance of [rehabilitation] exercises. (02-03)

### Subtheme 2.2: caregiver roles

Participants described postdischarge stroke care as largely the role of home-based caregivers (mainly family members) who had responsibility for a range of tasks, including rehabilitation, the prevention of complications and subsequent stroke, and the provision of psychosocial support. Caregivers supported activities of daily living such as maintaining hygiene, food preparation tailored to the patient's swallowing ability and dietary requirements (eg, diabetes, low salt, lower calorie); the preparation and administration of medication; rehabilitation (recommended by PT or aimed at improving speech and/or cognition); facilitating follow-up appointments; seeking help in an emergency; and supporting the patient's psychological/emotional needs.

They have to do the medication on time. They have to do the proper positioning, proper personal hygiene to the patient, and then they have to maintain the proper diet to the patient, to give nourishment to the patients. The entire role in the hospital, that has to be continued in the home also. (01-06)

### Subtheme 2.3: caregiver challenges

Hospital staff provided support and education to an identified caregiver in hospital in preparation for discharge, although these trained caregivers were not always the primary caregiver at home.

Sometimes in rural areas women do not come when their husband is admitted in the hospital and the caretaker is different, so all these things happen, and I feel very bad. (01-10)

Staff reported that caregivers encountered a multitude of challenges including: financial constraints in healthcare expenses and loss of income; a lack of time to provide care around work commitments; varying levels of literacy, ability to comprehend, and to adhere to instructions; difficulties transporting the patient to hospital appointments and so resorting to nearer sources of help (including from local healthcare centres (with or without trained professionals), informal help (pharmacists), complementary medicine (CM) (AUYSH) and unqualified professionals (faith-healers, quacks); and a dearth of community-based services. Challenges were particularly marked among rural populations.

One of the negative aspects of patients being in rural areas is that most caregivers go out to work in daytime… They are unable to give enough time for patient care and exercise. (02-03)

### Theme 3: patient and caregiver engagement post discharge

Provided at discharge, follow-up appointments were individualised and based on the patient's level of disability, comorbidities and available postdischarge support. In some sites, dedicated stroke helplines, Stroke Unit contact details or individual staff members' phone numbers were provided.

### Subtheme 3.1: postdischarge support—hospital

Staff reported that it was common for patients to have multiple follow-up appointments with their doctor, who was able to provide advice, treatment and referral to other services, as needed. Follow-up appointments usually took place at 1 month postdischarge but varied according to patient need. However, some patients had challenges in attending follow-up appointments, mainly due to the distance to the hospital from their home and travel costs.

The patient comes after one month or if the patient is outside of the area, he may come after two months, three months. That is up to the patient. That is up to their condition. That is up to their circumstances. (01-01)

For patients who were asked to, and who were able to, attend follow-up, referral to specialist support was made available if needed or if this was part of ongoing treatment.

Once they come to come back for follow-up in the OPD they are sent to physiotherapy OPD for review. (02-03)

Tele support was variable in availability and purpose. Staff reported that patients and caregivers were more likely to contact the hospital postdischarge when the patient had complex health needs (from their stroke and/or comorbidities). The most common reasons for seeking help through teleservices were problems with NG tubes (feeding and blockages); advice about preparing food in different consistencies and condition specific diets (low sugar, low salt); questions about medication (dosage, frequency and sideeffects);

catheter care; monitoring or changes in blood pressure; clarifications about PT-based exercises; and psychological problems (including anxiety around recurrent stroke and the impact of stroke on life expectancy).

> The patients are getting discharged, we give them our number. So, at this moment, we are trying to attend to them as much as possible with the phone. So, if they have any problem for example, if there is a catheter problem, or there is NG change we try to solve this through the phone. (02-05)

Some hospitals were able to refer patients with ongoing rehabilitation needs to rehabilitation clinics and networks; and where the patient lived some distance from the hospital to local healthcare facilities.

> There are three options. One is the inpatient programme, once they are medically stable. The second is to get admitted under a rehabilitation package. We have 7, 10 and 30-day packages where they stay in hospital and get rehabilitation. The next option is we have guest houses outside [the hospital] and they can stay there at very minimal prices and they then come on an OPD basis to the physiotherapist. The last option is our network of physios visit them at home or we send them videos through WhatsApp. (03-05)

At some hospitals, there were also postdischarge outreach facilities (including palliative care and community nurses), but at others there were no such resources.

> We have palliative care nurses who do a lot of home visits, twice weekly, if there is any patient who is bedridden and fully dependent, those patients, we connect them to the palliative care nurses. Palliative care nurses visit them at home and change NG tubes, catheters, dressings and many more of those things. (02-01)

> We arrange a particular community level therapist, and we have that contact number too, so they can simply contact us regarding the patient's issues. (06-06)

### Subtheme 3.2: postdischarge support—community based

Outside of hospital-based care, postdischarge support services (such as counselling/psychological support and physiotherapy) were not stroke specific and were considered by staff to deliver little additional benefit. Broader postdischarge support (outside hospital) was reportedly lacking in availability.

> There are some people who get people to massage [by private physiotherapists] and all the therapy techniques that were taught they will not use, they will use their own technique and do oil massage. So, it doesn't strengthen them, a weakness is there. (02-02)

Patients and caregivers, particularly in rural areas were often reported to have accessed CM, unqualified 'Doctors' or traditional faith healers post discharge.

> The community will support more after discharge, they usually do so at the time of rituals. Pujari [priest] will advise many things that are advantageous or disadvantageous. (05-01)

## DISCUSSION

To our knowledge, this is the first qualitative study to explore postdischarge stroke care with HCPs providing stroke services across India. This study identified a number of strengths including comprehensive individualised treatment planning, task shifting where there was a lack of specialists, comprehensive patient and caregiver training, and postdischarge care facilitated by hospital, and community-based services, where available. However, there were also some significant challenges.

Inpatient discharge care planning processes usually began soon after admission but were often not formally documented, a process which may have facilitated a more consistent and comprehensive approach. The provision of tailored and individualised information, education and training for patients and caregivers was often dependant on staff and resource availability, which commonly resulted in task shifting. Task-shifting describes a situation where a task is transferred to a HCP with a different or lower level of education and training, or to a person specifically trained to perform limited tasks.[24] As not all hospitals in the study had comprehensive MDTs, with reported shortages of dieticians, SWs and therapy staff, in these instances, nurses would often take on the roles of other HCPs despite limited formal training. While task shifting often enables a more efficient use of available human resources and can constitute a robust strategy to provide stroke care,[25] there is currently no documented evidence of stroke-specific task-shifting training for non-neurologist HCPs, with HCPs reporting that they largely developed knowledge and skills through informal or experiential learning.

Caregivers, often female family members, were responsible for all aspects of care and activities of daily living post discharge; results echoed in previous research.[26] Their ability to deliver required healthcare and psychosocial support was often dependent on access to services (geographically and financially), and the availability of other family members generating income in their (or the stroke survivors) absence. Congruent with other reports, HCPs identified challenges in training and education, and consequently difficulties for caregivers when stroke survivors had more severe strokes, requiring complex care.[27] While caregivers may have experienced increased demands, and in some cases limited preparedness for their new roles, unlike research from developed countries,[28] this was viewed by HCPs to be a necessary responsibility rather than a burden.

As postdischarge care is frequently the responsibility of family members in LMICs,[29] the preparation of caregivers was a vital aspect of in-hospital discharge planning. Training was usually delivered verbally, sometimes

supported by alternative formats provided in local languages to enhance accessibility, comprehension and adherence. However, patient and caregiver characteristics (demographic, socioeconomic, stroke severity) and social roles (employment status, family structure and dynamics), all had an impact on the subsequent accessibility, availability and provision of postdischarge care. Caregiver literacy was viewed as an importance aspect of successful postdischarge care. Previous research has demonstrated that low functional literacy may contribute to poor health status through a reduced ability to use written instructions and advice, and reduced adherence to medication and rehabilitation regimens.[30] Low literacy levels were viewed as a particular challenge in rural areas.

Postdischarge support provided by hospitals included multiple follow-up appointments, the frequency and number determined by the patient's condition. Postdischarge care included further service referrals (to rehabilitation clinics, local healthcare centres or postdischarge outreach facilities where available). However, postdischarge support beyond hospital settings was lacking in the community, as a result CM, unlicensed practitioners and faith healers were often sought, particularly among rural populations. In addition, other non-stroke-specific rehabilitation services with limited patient benefit were relied on. CM approaches for post-stroke recovery are frequently accessed in India and often include ayurvedic massage, herbal medicine, marma therapy, reiki therapy, homeopathy, intravenous fluids and opium.[31]

Further data and research are needed to understand the support available and needs of stroke survivors and caregivers, to inform the development of innovative post-stroke interventions that are accessible, patient-centred and culturally sensitive.[18]

This study has a number of limitations. First, the study did not seek to interview HCPs based in the community as these services varied between the included sites and were not stroke specific. Due to the diversity of the included hospitals, we were unable to make comparisons across services. Hospital staff broadly lacked awareness of the spectrum of stroke care services available in-hospital (outside of their own profession) and in the community. Finally, it is acknowledged that one in every three patients who had a stroke do not access appropriate healthcare due to non-affordability, usage of alternative medicines and difficulty in conveyance to hospital.[9]

We have also identified several strengths: participants represented a broad range of stroke services; of which, over a third were nurses of varying seniority. Interviews were conducted in local languages to maximise engagement and by providing participants with a choice of interview method (telephonic or face-to-face), this enabled adherence to COVID-19 protocols, and supported inclusion of participants under COVID-19 restrictions at the point of interview.

## CONCLUSION

Discharge planning was integrated and customised but in some sites there were limited resources. Task shifting among HCPs compensated for a lack of specialists, but limitations were identified in staff education and training. Caregivers faced many challenges in both accessing and providing postdischarge care. Postdischarge care was mainly hospital based, supported by teleservices, especially for rural populations. Further research is needed to gain an understanding of postdischarge care provision and the needs of stroke survivors and their caregivers.

**Author affiliations**
[1]School of Nursing and Midwifery, University of Central Lancashire, Preston, UK
[2]Department of Neurology, Christian Medical College and Hospital Ludhiana, Ludhiana, Punjab, India
[3]Department of Neurology, Sree Chitra Tirunal Institute for Medical Sciences and Technology, Thiruvananthapuram, Kerala, India
[4]Department of Neurology, All India Institute of Medical Sciences, New Delhi, India
[5]Department of Neurology, Indira Gandhi Medical College, Shimla, Himachal Pradesh, India
[6]Department of Medicine, Baptist Christian Hospital Tezpur, Tezpur, Assam, India
[7]Department of Neurology, National Institute of Mental Health and Neuro Sciences, Bangalore, Karnataka, India
[8]Department of Neurology, Zydus Research Center, Ahmedabad, Gujarat, India

**Acknowledgements** We would like to thank the health professionals who took the time to take part in this study. We would also like to thank the members of the NIHR Global Health Research Group on IMPROVIng Stroke CarE in India (IMPROVISE) Collaboration.

**Collaborators** NIHR Global Health Research Group on IMPROVIng Stroke CarE in India (IMPROVISE) Collaboration. Dr. Jeyaraj D Pandian Principal and Professor, Department of Neurology Christian Medical College and Hospital, Ludhiana, India Email: jeyarajpandian@hotmail.com Dr. M.V. Padma Srivastava Professor of Neurology, All India Institute of Medical Sciences, Delhi, India Email: vasanthapadma123@gmail.com Dr. P.N. Sylaja Professor of Neurology, Sree Chitra Tirunal Institute of Medical Sciences and Technology, Thiruvananthapuram, Kerala, India Email: sylajapn@hotmail.com Professor Dame Caroline Watkins Professor of Stroke & Older People's Care / Faculty Director of Research and Innovation, University of Central Lancashire, Preston, United Kingdom Email: clwatkins@uclan. ac.uk Dr Liz Boaden Senior Research Fellow, University of Central Lancashire, Preston, United Kingdom Email: EBoaden1@uclan.ac.uk Associate Professor Dominique Cadilhac Head of Translational Public Health Research Stroke and Ageing Research Group, Monash University, Clayton VIC 3168, Australia Email: Melbournedominique.cadilhac@monash.edu Professor Andy Clegg Professor Health Services Research, University of Central Lancashire, Preston, United Kingdom Email: AClegg3@uclan.ac.uk Mrs Denise Forshaw Principal Clinical Trials Manager and Deputy Director (Operations and Governance, Lancashire Clinical Trials Unit, University of Central Lancashire, Preston, United Kingdom Email: Dforshaw@ uclan.ac.uk Professor Mark Gabbay Director of NIHR NWC CLAHRC, Primary Care and Health inequalities, University of Liverpool, United Kingdom Email: M.B. Gabbay@liverpool.ac.uk Ms Rachel GeorgiouSenior Research Fellow, University of Central Lancashire, Preston, United Kingdom Faculty of Health & Wellbeing, School of Nursing, Email: RGeorgiou@uclan.ac.uk Dr Jo GibsonReader in Applied Health Research, University of Central Lancashire, Preston, United Kingdom Email: JGibson4@uclan.ac.uk Professor Maree Hackett Program Head Mental Health, The George Institute for Global Health, Neurological & Mental Health Division, NSW 2050, Australia Email: mhackett@georgeinstitute.org.au Dr Steph Jones Programme Manager and Reader in Applied Health Research, University of Central Lancashire, Preston, United Kingdom Email: sjones10@uclan.ac.uk Dr Liz Lightbody Professor of Stroke Care and Improvement, University of Central Lancashire, Preston, United Kingdom Email: CELightbody@uclan.ac.uk Professor Sandy Middleton Director, Nursing Research Institute, Australian Catholic University & St Vincent's Health Australia, NSW 2010, Australia Email: Sandy.Middleton@acu. edu.au Dr Gordon Prescott Reader in Medical Statistics and Deputy Director of the Lancashire Clinical Trials Unit, University of Central Lancashire, Preston, United Kingdom Email: GPrescott1@uclan.ac.uk Professor Caroline Sanders Professor of Medical Sociology, University of Manchester, Manchester, United Kingdom Email: Caroline.Sanders@manchester.ac.uk Dr Anil Sharma Clinical adviser to NIHR Global Health Research Group, University of Central Lancashire, Preston, United Kingdom Email: anilamrala@gmail.com Professor Marion Walker Professor of Stroke Rehabilitation; a.) Associate Pro Vice-Chancellor (Equality, Diversity and

Inclusion) Nottingham University, United Kingdom b.) Department of Health through the National Institute for Health Research, United Kingdom Email: marion.walker@nottingham.ac.uk.

**Contributors** CEL, MVPS, JDP, PNS and CLW contributed to the study conception and study design. Data collection was undertaken by PK, SR, SS, AK, AD, JP, VB. Analysis was performed by SPJ, RJI, JCW, PK, SR, SS, AK, AD, JP, VB, JDP, CEL, MVPS, JDP, PNS, CLW, JJW, AS, SS, and GBK. The first draft of the manuscript was written by SPJ, RJI and JCW. All authors commented on previous versions of the manuscript. All authors read and approved the final manuscript. SPJ acts as guarantor for this manuscript.

**Funding** This research was funded by the National Institute for Health Research (NIHR) Global Health Research Groups.

**Disclaimer** This research was funded by the National Institute for Health Research (NIHR) (Global Health Research Group on Improving Stroke Care in India, University of Central Lancashire (16/137/16)) using UK aid from the UK Government to support global health research. The views expressed in this publication are those of the author(s) and not necessarily those of the NIHR or the UK government.

**Competing interests** None declared.

**Patient and public involvement** Patients and/or the public were involved in the design, or conduct, or reporting, or dissemination plans of this research. Refer to the Methods section for further details.

**Patient consent for publication** Not applicable.

**Ethics approval** This study involves human participants and was approved by Ethics approval (REF: 2020-9715) was granted by the Health Ministry's Screening Committee (HMSC), India and each hospital's ethics committee. The study conforms to ethical standards of the Indian Council of Medical Research Guidelines. In the UK, ethical approval was granted by the Science, Technology, Engineering and Medicine Ethics Committee, University of Central Lancashire (STEM 939). Informed consent was obtained from all participants prior to interview. Participants gave informed consent to participate in the study before taking part.

**Provenance and peer review** Not commissioned; externally peer reviewed.

**Data availability statement** Data are available upon reasonable request. Not applicable.

**ORCID iDs**
Stephanie P Jones http://orcid.org/0000-0001-9149-8606
Ranjit J Injety http://orcid.org/0000-0003-1403-8336
Jeyaraj D Pandian http://orcid.org/0000-0003-0028-1968
Jemin Webster http://orcid.org/0000-0002-2506-7135
Jo Catherine Weldon http://orcid.org/0000-0003-0729-8121

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
