## [Reviewer comments · BMJ Open]

ARTICLE DETAILS

TITLE (PROVISIONAL)	Healthcare professionals' perspectives of the provision of, and challenges for, eating, drinking and psychological support post-stroke: findings from semi-structured interviews across India.
AUTHORS	Jones, Stephanie; Injety, Ranjit; Pandian, Jeyaraj; Ratra, Sanjali; Sylaja, PN; Babu, Veena; Srivastava, Padma; Sharma, Sakshi; Sharma, Sudhir; Webster, Jemin; Koirala, Amrit; Kaushal, Pawna; Kulkarni, Girish Baburao; Dixit, Anand; Sharma, Arvind; Prajapati, Jagruti; Weldon, Jo; Kuroski, Jennifer A; Watkins, Caroline; Lightbody, Catherine

VERSION 1 – REVIEW

REVIEWER	Horgan, Frances RCSI
REVIEW RETURNED	04-Dec-2022

GENERAL COMMENTS	Very interesting topic. The title does not reflect the focus on discharge processes and supports for swallow, eating and psychological support as outlined on page 6, lines 18-24 methods. The statement at the start of the methods states the following ' Interviews explored HCPs' views of stroke patient management in preparation for discharge and post discharge support for patients and caregivers, focussing on eating, drinking and psychological support'. If the focus was on supports for focussing on eating, drinking and psychological support, this should be reflected in the title, abstract and aim of the study. Questions 4 and 5 in the thematic interview guide also mentioned eating, drinking and psychological support. Background - there's limited detail on the background and context for the study and current services in India, there is some text in the discussion that could be included in the background section to provide this context. See text on page 15 discussion lines 8-20, the paragraph starts with 'The resources for post stroke...' The term task shifting is mentioned throughout the manuscript, what does this term mean, add a definition / explanation. Methods Why were 66 interviews conducted, what was the rationale for including 7 hospitals? How were the 7 hospitals selected, were they a similar type of hospital?
---

	How was data saturation not reached before 66 interviews were completed? In the methods section the authors state in the data collection section, page 6 line 42-44 that 'data collection continued until saturation'. Please provide some additional details on how this was determined. What methods was used for the data analysis, was it reflexive thematic analysis? Results, sub theme 1.1 Healthcare system and resources, the opening statement is as follows: 'all sites had localised, inpatient discharge planning processes... this seems to contradict the quotation where the interviewee (01-05) stated that 'we don't have written protocols ? Results Theme 3, page 12 line 39-43, when were the followup appointments? Page 13 in the first and second paragraphs, the term, ' task shifting' is used, what does this mean? The strengths and limitations of the study are quite limited in detail at the end of the discussion, there is a more detailed table. review for consistency of details presented. Add page numbers to the COREQ checklist. Is there a table to profile the characteristics of the interviewees? The authors include the study protocol, WP 1 2 3 as outlined in the study protocol includes interviews with 10 patients and relatives, is this a separate study / will be reported separately. Clarify any deviations from the original study protocols registered.
--	--

REVIEWER	Forrest, George Albany Medical College, PM&R
REVIEW RETURNED	21-Dec-2022

GENERAL COMMENTS	This is not an organized research study that collects information in a systematic manner and analyzes data using appropriate statistical means. It is more appropriate for a newspaper or magazine than for a scientific journal. There also is no reason to think that the opinions offered generalize to a different setting other than the region of India where it was conducted
--

VERSION 1 – AUTHOR RESPONSE

Reviewer 1 What methods was used for the data analysis, was it reflexive thematic analysis?	Included page 6 - Patients and the public were involved in discussions to agree the priority areas that were the focus of post-discharge support interviews with healthcare professionals (eating, drinking and psychological support).
--	--

Results, sub theme 1.1 Healthcare system and resources, the opening statement is as follows: 'all sites had localised, inpatient discharge planning processes... this seems to contradict the quotation where the interviewee (01-05) stated that 'we don't have written protocols ?	Yes, reflective thematic analysis was the method used. Added to the text in the analysis section on page 6 - Each transcript was read in full, and data explored using inductive coding and analysed using a reflexive thematic method.
Results Theme 3, page 12 line 39-43, when were the follow-up appointments?	Extra text added to the quote on page 8 - "We have discharge processes that the team work to but we don't have any written protocol but we are observing the patient from the admission." (01-05)
Page 13 in the first and second paragraphs, the term, ' task shifting' is used, what does this mean?	Extra text added on page 12 - Follow-up appointments usually took place at one-month post-discharge but varied according to patient need. Extra text added to explain on page 14 - task-shifting (defined as the rational redistribution of tasks among health workforce teams).
The strengths and limitations of the study are quite limited in detail at the end of the discussion, there is a more detailed table. review for consistency of details presented.	Extra text added on page 16 and made consistent with the detailed table earlier in the manuscript.
Add page numbers to the COREQ checklist.	Updated to reflect changes and page numbers added to COREQ checklist.
Is there a table to profile the characteristics of the interviewees?	Table added on page 7.
The authors include the study protocol, WP 1 2 3 as outlined in the study protocol includes interviews with 10 patients and relatives, is this a separate study / will be reported separately.	Yes, this is a separate study and will be reported separately.
Clarify any deviations from the original study protocols registered.	None to report.

Reviewer 2 comments:

This is not an organized research study that collects information in a systematic manner and analyzes data using appropriate statistical means. It is more appropriate for a newspaper or magazine than for a scientific journal. There also is no reason to think that the opinions offered generalize to a different setting other than the region of India where it was conducted.

Thank you for your comments. We believe this study has utilised appropriate qualitative methods and due to the diversity of the 7 hospitals in the study, findings may be generalisable to other similar settings in India.